# Changes in apparent consumption of staple food in Mexico associated with the gradual implementation of the NAFTA

**Néstor A. Sánchez-Ortiz[1,2], Mishel Unar-Munguía[1], Sergio Bautista-Arredondo[2], Teresa Shamah-Levy[3], M. Arantxa Colchero[2]***

**1** Center for Research on Nutrition and Health, National Institute of Public Health, Cuernavaca, Morelos, Mexico, **2** Center for Health Systems Research, National Institute of Public Health, Cuernavaca, Morelos, Mexico, **3** Center for Evaluation and Surveys Research, National Institute of Public Health, Cuernavaca, Morelos, Mexico

* acolchero@insp.mx

**Data Availability Statement:** The data used for this study are publicly available from FAOSTAT (https://www.fao.org/faostat/en/#data/FBSH) [outcome

## Abstract

In 1994, the United States, Canada, and Mexico signed the North American Free Trade Agreement (NAFTA) to remove trade barriers and facilitate cross-border trade in goods and services. Worldwide, trade agreements, urbanization and economic development have shaped significant changes in dietary habits. This study aims to evaluate the association between the gradual implementation of NAFTA and changes in apparent consumption of staple foods in Mexico. We analyzed national apparent consumption of animal- and vegetable-source foods, using data from the Food and Agriculture Organization of the United Nations (FAO) from 1970 to 2018. Association between NAFTA and apparent consumption was estimated using interrupted time series analysis (ITSA) with synthetic controls and included two inflection points based on the implementation of NAFTA: 1994, when the agreement began, and 2008 when it was fully implemented. As a result, comparing Mexico with the synthetic control, we found a significant decrease in apparent consumption of pulses, -3.22 and -1.92 kcal/capita/day in the post-1994 and post-2008 periods, respectively. The vegetable-source foods showed an increase of 5.79 kcal/capita/day after 2008. The trends of apparent consumption of animal-source foods, eggs, and milk had significant increases after 1994 and 2008. The apparent consumption of meat increased only after 2008. The implementation of NAFTA was associated with an increase in apparent consumption of food from animal-source and a decrease in consumption of pulses. After 2008, an increase in apparent consumption of vegetable-source foods was observed.

## Introduction

Globally, urbanization and economic development have been accompanied by significant changes in dietary patterns [1]. Traditional diets, such as the Mexican diet, based on a limited number of staple foods, have changed to an increased proportion of animal-source foods (meat, milk, and eggs), vegetable oils, and sugars. This process is known as "the nutrition

variable: apparent consumption of staple foods (FAO- Food Balance Sheets)] and World Bank (https://data.worldbank.org/country/mexico?view=chart) [predictor variables: gross domestic product, life expectancy at birth, Co2 emissions per capita, the proportion of the rural population, and primary school enrollment] databases.

**Funding:** This work was funded by Bloomberg Philanthropies awarded to MAC. This work was also supported by the National Council of Science and Technology of Mexico (CONACyT) in the form of a scholarship grant (547311) awarded to NASO. The funders had no role in study design, data collection and analysis, decision to publish, or preparation of the manuscript.

**Competing interests:** The authors have declared that no competing interests exist.

transition" [2–4], and it has been associated to changes in employment, urbanization; macro-economic causes such as economic reforms; and the introduction of new technologies (for food production and processing) [4–6]. Among the most important economic causes of the "nutrition transition" is the global integration of markets through free trade agreements, which remove all or some trade barriers [7,8]. Trade liberalization affects the availability and the retail prices by lowering or eliminating tariffs of imports, facilitating foreign investment in the food industry, processing, retailing and advertising, and encouraging the growth of trans-national food companies [9]. Trade agreements promote monoculture agriculture (e.g., sugar, corn, soybean) to produce ultra-processed products [10,11]. These changes affect local food consumption patterns [1,12,13].

Removing trade barriers and increasing investment incentives facilitate transnational food corporations to buy small companies, products, and services across borders [4]. The result is that a single company takes over the production, distribution, and sales of a particular food, promoting changes in the food system that encourage the consumption of processed foods [14].

Some studies have analyzed the relationship between free trade agreements and changes in food consumption. In Thailand, the integration of the Association of Southeast Asian Nations (ASEAN) Free Trade Area has been associated with an increase in snack food consumption [1]. In Costa Rica, El Salvador, Guatemala, Honduras, and Nicaragua, the implementation of the Central America-Dominican Republic Free Trade Agreement (CAFTA-DR) was followed by an increase in the consumption of meat and processed foods [12,15]. In Vietnam and Peru, the implementation of regional trade agreements yielded increased consumption of sweetened beverages [16,17].

The North American Free Trade Agreement (NAFTA) was signed in 1994, which aimed to remove trade barriers and facilitate cross-border trade in goods and services between the United States, Canada, and Mexico, over the following 15-years [18,19]. NAFTA successfully increased investment opportunities in the territories of the parties involved [19]. In the food industry alone, investment in Mexico from the United States increased from $2.3 to $8.7 billion between 1993 and 2007 [18]. However, during the NAFTA implementation period, Mexico experienced an increase in type two diabetes [20], overweight and obesity [19].

The World Trade Organization claims that health protection is a significant concern, and therefore trade agreements must seek a balance between "trade and health" [21]. There is evidence that NAFTA has promoted unhealthy foods such as processed meat or sweeteners [22–24], but to our knowledge, the relationship between the gradual implementation of NAFTA and changes in apparent consumption of other staple foods has not yet been explored. To isolate the potential effect of NAFTA, we evaluated changes in apparent consumption of staple foods divided by animal- and vegetable-source food between Mexico and a synthetic control (constructed with information from a pool of lower-middle and upper-middle income countries) before and after two inflection periods: 1994, when the agreement came into force, and 2008, when NAFTA was fully implemented.

## Methods

### Data sources

We analyzed apparent consumption data from the Food Balance Sheets (FBS) of the Food and Agriculture Organization of the United Nations (FAO), for Mexico and a group of 55 lower-middle and upper-middle-income countries between 1970 and 2018. Apparent consumption is defined as the average supply of food and nutrients, expressed in grams or calories, by individual or grouped food items, available for consumption, and divided by the country's

population in the middle of a given year [25]. The FBS estimates apparent national consumption of the main staple food groups and used nutritional factors to convert kilograms of staple food groups into calories [26]. We divided annual calories by 365 days to determine daily calories by food group.

For this study, available information on eighteen food groups, were classified into two major categories: animal and vegetable, depending on their origin. Animal foods includes meat, offal, animal fats, eggs, milk, fish, and seafood. We also defined the group of animal-source food products as the sum of meat, offal, eggs, and milk. Vegetable foods includes starchy roots, cereals, sugar plants, pulses, tree nuts, vegetable oils, vegetables, and fruits. In this group, we included two additional food categories: vegetable-source food, which encompasses all vegetable-origin foods, and fruits and vegetables, which include only pulses, vegetables, and fruits.

## Statistical analysis

**Interrupted time series analysis with synthetic controls.** In experimental designs where treatment and control groups are randomly assigned, comparability between them at baseline in basic characteristics is guaranteed. In the absence of such a design, we rely on statistical tests to check for comparability. When the baseline magnitude and trends in the outcomes of interest (kcal/capita/day for each food group) between a control and treatment group are not statistically different in the pre-intervention period, we expect that in the absence of a shock, such as NAFTA, the parallel trends would continue. Therefore, the difference in the outcomes between control and treatment groups after implementation can be attributed to the intervention. If the trends are not comparable, for instance, if the pre-intervention trend is steeper for the treatment group, the post-intervention difference would be upward biased.

We used interrupted time series analysis (ITSA) with synthetic controls to estimate the association between NAFTA and changes in apparent staple food consumption in Mexico [27–29]. ITSA is a method for assessing the effectiveness of large-scale interventions at the aggregate level [27] when there is no control group from an experimental design and the intervention is expected to change the trend of outcome variables [30]. The ITSA assesses the comparability of the study groups with two parameters: the intercept and the average change for the pre-intervention period. The addition of a synthetic control group is roughly equivalent to including treated and nontreated units, that were similar in observed characteristics prior to the intervention. Therefore, the addition of a control group strengthens the ITSA design [27].

The synthetic controls method predicts outcomes from a hypothetical control group based on a weighted average of preselected covariates from a group of nontreated units (donor pool) [31]. We included a list of lower-middle and upper-middle-income countries for which complete information on all variables was available. We excluded all countries that had bilateral trade agreements with the United States or Canada during the period analyzed. We obtained this information from World Bank databases [32].

The weights given to each donor unit in the construction of synthetic control are determined by the similarity between the observed characteristics of the countries in the donor pool and Mexico. Observations from countries that match better receive a higher weight, with values between 0 and 1 [33]. To estimate the effect of NAFTA, the synthetic control model predicts trends in apparent consumption of staple foods in a synthetic country that is comparable to Mexico but does not implement NAFTA. We chose the following variables to predict staple food consumption: a) apparent food consumption (in calories/capita/day) and supply of fat and protein expressed in gr/capita/day provided by the modeled food group; b) gross domestic product, life expectancy at birth, Co2 emissions per capita (as a proxy for industrial and other

economic activities), the proportion of the rural population, and primary school enrollment. School enrollment is a rate for the total number of students enrolled in primary education, calculated as a percentage of the total population of official primary school age. This indicator may be higher than 100% due to the inclusion of students older or younger than the official age [32].

**Association between the gradual implementation of NAFTA with changes in apparent consumption of staple foods.** Studies have used this method to estimate the effect of NAFTA on the consumption of sweeteners, therefore we adapted the methodology to estimate changes in apparent consumption of staple foods [22,24]. To evaluate the association between the gradual implementation of NAFTA and changes in apparent consumption of staple foods, we modeled three splines: before the NAFTA implementation from 1970 to 1993, from the beginning of the agreement in 1994 to 2008, when it was fully implemented, and after 2008. We estimated an ITSA model as follows:

$$Y_t = \beta_0 + \beta_1 T_t + \beta_{2-3} X_{i,t} + \beta_{4-5} X_{i,t} T_t + \beta_6 D + \beta_7 D T_t + \beta_{8-9} D X_{i,t} + \beta_{10-11} D X_t T_t + \varepsilon t$$

Where $Y_t$ represents the apparent consumption of food groups (outcome variable) at time $t$, $T_t$ is a count variable representing the years from 1970 to 2018, $X_{i,t}$ are two dummy variables (pre-intervention period = 0 and intervention = 1) for each spline, and $D$ is a dummy variable indicating the intervention or synthetic control group. The coefficient $\beta_0$ denotes the intercept for the control group, $\beta_1$ represents the pre-intervention trend for the control group, $\beta_{2-3}$ are the change in the outcome level for the control group in each spline after the intervention, and $\beta_{4-5}$ shows the change in the outcome trend in each spline for the control group.

A strength of ITSA models with treatment and control units is that we can assess comparability between study groups in the covariates. The parameters $\beta_6$ and $\beta_7$ indicate whether the treatment and control units are balanced in the outcome variable's level and trend during the pre-intervention period. In the context of an experimental design, we would not expect to find differences [27]. It also tests for differences in level between the intervention and control units in each intervention splines points (1994 and 2008), representing the outcome level immediately after the interventions happens ($\beta_{8-9}$). Finally, it tests for differences in the trends between the intervention and control groups in the post-interventions period, compared with the pre-intervention period, which is similar to a difference in difference estimation ($\beta_{10-11}$). The $\varepsilon t$ represents the error term.

## Results

### ITSA with synthetic control

We generated 18 synthetic control units, one for each food group. All controls were balanced with Mexico regarding the variables defined in the methods section (S1 Table shows the weighted values of each donor unit for the synthetic control and S2 Table presents the predictor variables for Mexico and the synthetic control). Table 1 shows the mean values of the outcome and predicting variables for staple food consumption for Mexico and the synthetic control. Once the country selection was completed, fifty-five units were available to form the donor pool used to construct the synthetic control.

Table 2 shows the coefficients for the intercept and pre-intervention trend to see if Mexico and the control groups are comparable. The ITSA model test for differences in the intercept (difference in the apparent consumption of the staple food group at baseline -1970-) between Mexico and the synthetic control; and, the pre-intervention trend that is the average change per year in the apparent consumption of a staple food before 1994 in Mexico compared to the synthetic control. After assessing comparability, only eight of the eighteen groups showed

**Table 1. Apparent consumption of food energy, nutrients, demographic, and economic characteristics for Mexico and donor pool countries in the pre-intervention period, from 1970 to 1993.**

| Variable | Mexico (n = 25 years) | | Donor Pool (n = 1350 years) | |
|---|---|---|---|---|
| | Mean | (SD) | Mean | (SD) |
| **Outcomes (Kcal/capita/day)** | | | | |
| Vegetable-source food | 2430.21 | (141.94) | 2050.20 | (344.06) |
| Fruits and vegetables | 268.46 | (38.30) | 181.13 | (91.29) |
| Cereals | 1415.50 | (41.23) | 1129.23 | (385.44) |
| Oilseeds | 22.50 | (6.11) | 78.70 | (125.46) |
| Starchy Roots | 23.58 | (2.28) | 191.01 | (243.63) |
| Sugar and Sweeteners | 424.46 | (43.18) | 236.47 | (136.77) |
| Fruits | 94.92 | (9.72) | 95.48 | (73.95) |
| Vegetables | 26.04 | (5.86) | 29.42 | (22.70) |
| Pulses | 147.50 | (35.34) | 55.26 | (45.44) |
| Vegetable Oils | 211.46 | (53.31) | 167.63 | (100.35) |
| Nuts | 4.63 | (0.92) | 5.67 | (10.04) |
| Animal-source food | 385.38 | (51.71) | 255.84 | (173.56) |
| Meat | 194.88 | (34.76) | 138.34 | (118.54) |
| Eggs | 30.21 | (7.99) | 12.96 | (11.24) |
| Milk | 148.63 | (16.95) | 98.47 | (80.44) |
| Fish and seafood | 14.38 | (5.75) | 28.08 | (28.28) |
| Offal | 11.67 | (2.58) | 6.51 | (5.33) |
| Animal Fats | 52.25 | (17.36) | 47.57 | (48.74) |
| **Predictors** | | | | |
| Food supply (kcal/capita/day) | 2882.13 | (201.78) | 2367.42 | (409.98) |
| Protein supply quantity (g/capita/day) | 76.26 | (6.14) | 61.09 | (14.78) |
| Fat supply quantity (g/capita/day) | 73.60 | (10.48) | 56.46 | (22.48) |
| *Gross domestic product (million dollars) | $169,000.00 | ($113,000.00) | $28,600.00 | ($58,700.00) |
| life expectancy at birth (years) | 67.00 | (3.31) | 60.12 | (8.31) |
| Co2 emissions (metric tons per capita) | 3.46 | (0.65) | 1.56 | (1.91) |
| School enrollment, primary (%) | 114.32 | (5.77) | 95.76 | (23.99) |
| Rural population (%) | 33.40 | (4.20) | 63.00 | (17.98) |

*Deflated values in United States dollars for 2018. Number of observations is based on 25 years per country. Counties included in donor pool: Albania, Algeria, Arab Republic, Argentina, Belize, Benin, Bolivia, Botswana, Bulgaria, Cambodia, Cameroon, Cape Verde, China, Congo Republic, Côte d'Ivoire, Cuba, Djibouti, Ecuador, Egypt, Eswatini, Fiji, Ghana, Grenada, Guyana, India, Indonesia, Iran, Islamic Republic, Jamaica, Kenya, Kiribati, Lao, Malaysia, Mauritania, Mongolia, Nepal, Nigeria, Pakistan, Paraguay, Philippines, Saint Lucia, Saint Vincent and the Grenadines, Samoa, Senegal, Solomon Islands, South Africa, Sri Lanka, Suriname, Thailand, Tunisia, Turkey, Vanuatu, Venezuela, Viet Nam, Zambia, Zimbabwe.

parallel trends in the pre-intervention period so were kept for subsequent analyzes. Selected groups were: vegetable-source foods, fruits and vegetables, pulses, nuts, animal-source foods, meat, eggs, and milk (Table 2). Fig 1 shows the graphical representation of the observed and model-predicted apparent consumption trends in Mexico and the synthetic control.

## Staple food consumption

Table 3 shows NAFTA's effect estimations on staple food consumption in Mexico. Post-1994 trend shows no statistically significant changes in apparent consumption of vegetable-source food. However, the post-2008 trend shows a significant increase of 5.05 kcal/capita/day (p = 0.02). The synthetic control's slope coefficient was negative but not statistically significant;

**Table 2. Comparability between Mexico and the synthetic control by food group: Differences in the intercept and pre-intervention trend (1970–1993).**

| | Difference (Kcal/capita/day) | p-value | 95% CI | | Difference (Kcal/capita/day) | p-value | 95% CI |
|---|---|---|---|---|---|---|---|
| **Vegetable-source food** | | | | **Vegetable Oils** | | | |
| Intercept | 53.49 | 0.080 | [-6.77, 113.75] | Intercept | -49.5 | 0.000 | [-65.87, -33.14] |
| pre-intervention trend | 3.13 | 0.160 | [-1.25, 7.52] | pre-intervention trend | 3.98 | 0.000 | [2.62, 5.33] |
| **Fruits and vegetables** | | | | **Nuts** | | | |
| Intercept | -25.13 | 0.070 | [-51.89, 1.64] | Intercept | -0.31 | 0.440 | [-1.12, 0.49] |
| pre-intervention trend | -0.53 | 0.520 | [-2.18, 1.11] | pre-intervention trend | -0.03 | 0.490 | [-0.11, 0.05] |
| **Cereals** | | | | **Animal-source food** | | | |
| Intercept | 111.47 | 0.000 | [81.81, 141.14] | Intercept | 15.95 | 0.480 | [-28.33, 60.23] |
| pre-intervention trend | -5.94 | 0.000 | [-8.58, -3.30] | pre-intervention trend | -0.41 | 0.780 | [-3.36, 2.54] |
| **Oilseeds** | | | | **Meat** | | | |
| Intercept | 5.25 | 0.010 | [1.61, 8.89] | Intercept | 6.5 | 0.620 | [-19.27, 32.27] |
| pre-intervention trend | -0.59 | 0.000 | [-0.83, -0.35] | pre-intervention trend | -0.63 | 0.480 | [-2.41, 1.14] |
| **Starchy Roots** | | | | **Eggs** | | | |
| Intercept | -3.97 | 0.050 | [-7.95, 0.01] | Intercept | 1.34 | 0.200 | [-0.74, 3.41] |
| pre-intervention trend | -0.89 | 0.000 | [-1.28, -0.51] | pre-intervention trend | 0.02 | 0.830 | [-0.17, 0.21] |
| **Sugar and Sweeteners** | | | | **Milk** | | | |
| Intercept | 52.67 | 0.000 | [26.99, 78.35] | Intercept | -6.84 | 0.470 | [-25.52, 11.84] |
| pre-intervention trend | 4.88 | 0.000 | [3.24, 6.52] | pre-intervention trend | 0.28 | 0.650 | [-0.95, 1.52] |
| **Fruits** | | | | **Fish and seafood** | | | |
| Intercept | -51.29 | 0.000 | [-59.18, -43.39] | Intercept | -5.08 | 0.000 | [-7.66, -2.50] |
| pre-intervention trend | 2.13 | 0.000 | [1.56, 2.70] | pre-intervention trend | 0.41 | 0.000 | [0.24, 0.58] |
| **Vegetables** | | | | **Offal** | | | |
| Intercept | -3.83 | 0.010 | [-6.62, -1.03] | Intercept | -5.24 | 0.000 | [-5.90, -4.59] |
| pre-intervention trend | 0.42 | 0.000 | [0.21, 0.62] | pre-intervention trend | 0.42 | 0.000 | [0.38, 0.47] |
| **Pulses** | | | | **Animal Fats** | | | |
| Intercept | 15.86 | 0.210 | [-9.15, 40.87] | Intercept | -19.5 | 0.000 | [-23.00, -16.01] |
| pre-intervention trend | 0.1 | 0.890 | [-1.30, 1.50] | pre-intervention trend | 1.56 | 0.000 | [1.21, 1.92] |

The intercept is the difference in the apparent consumption of the staple food group at baseline (1970) between Mexico and the synthetic control. The pre-intervention trend is the average change per year in the apparent consumption of a stable food before 1994 in Mexico compared to the synthetic control. Only food groups with non-significant differences (p>0.05) between Mexico and the control group in the two parameters were considered comparable. 95%CI—95% confidence interval.

the difference between the slopes was 5.79 kcal/capita/day (p = 0.02), representing a significant increase in the apparent consumption of vegetable-source foods in Mexico after NAFTA was fully implemented.

No changes in trends were seen in the fruit and vegetable groups in Mexico. Regarding the synthetic control, we saw increases of 1.5 kcal/capita/day in 1994 and 2.03 kcal/capita/day in 2008. However, no differences were found between treatment and control groups. Regarding pulses, after 1994, Mexico's estimated trend was -1.70 while the control's trend was 1.52 kcal/capita/day (p < 0.05). The direction of the coefficients remained the same after 2008. However, it was significant only for the control, and the difference was -1.92 kcal/capita/day (p = 0.01), indicating that Mexico experienced a decline in apparent consumption of pulses after the full release of NAFTA.

During the first phase of NAFTA (1994–2007), both Mexico and the synthetic control experienced an increase in apparent consumption of nuts, with values of 1.16 (p = 0.00) and 0.14 (p = 0.01) kcal/capita/day. The difference between trends implies an increase of 1.01 (p = 0.00)

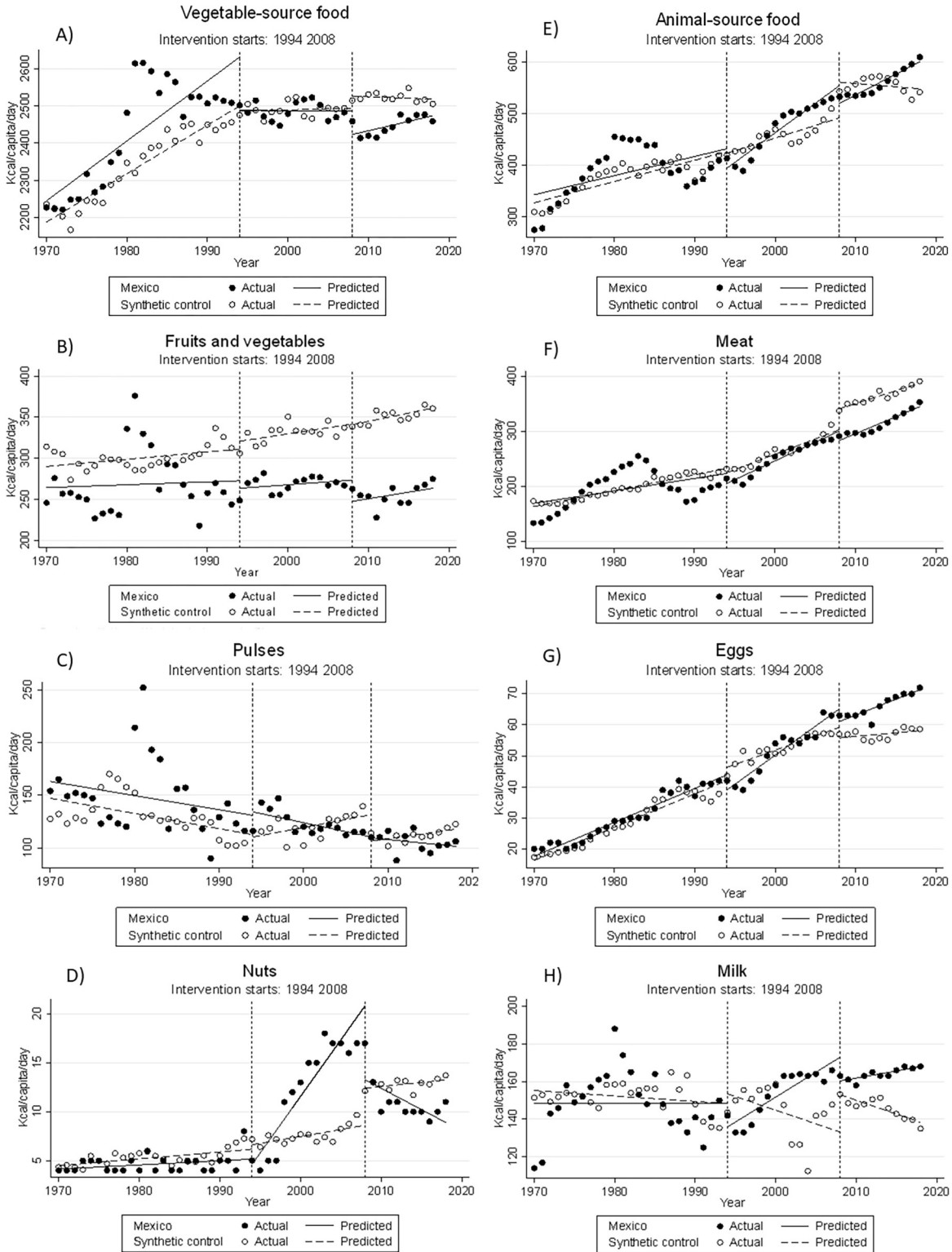

**Fig 1. Observed and predicted values from the ITSA regression model before and after NAFTA implementation, comparing Mexico and the synthetic control (kcal/capita/day by food group).** A) Vegetable-source food; B) Fruits and vegetables; C) Pulses D) Nuts; E) Animal-source food; F) Meat; G) Eggs; H) Milk. The control group was estimated by synthetic controls from a pool of lower-middle and upper-middle-income countries, adjusted for supply of fat and protein (gr/capita/day) provided by the modeled food group, gross domestic product, life expectancy at birth, $CO_2$ emissions per capita, the proportion of the rural population, and primary school enrollment.

**Table 3. Post 1994 and 2008 intervention trends of stable food apparent consumption (kcal/capita/day), comparing Mexico and the synthetic control group.**

| | | 1994-initial implementation | | | 2008- full implementation | | |
|---|---|---|---|---|---|---|---|
| | | Coefficient | p-value | 95% CI | Coefficient | p-value | 95% CI |
| **Vegetable-source food** | Mexico | -0.19 | 0.880 | [-2.70, 2.31] | 5.05 | 0.020 | [0.84, 9.26] |
| | Synthetic control | 0.64 | 0.440 | [-1.01, 2.28] | -0.74 | 0.510 | [-2.95, 1.48] |
| | Difference | -0.83 | 0.580 | [-3.82, 2.17] | 5.79 | 0.020 | [1.04, 10.55] |
| **Fruits and vegetables** | Mexico | 0.71 | 0.270 | [-0.57, 1.99] | 1.58 | 0.160 | [-0.66, 3.82] |
| | Synthetic control | 1.50 | 0.030 | [0.14, 2.86] | 2.03 | 0.000 | [3.71, 7.83] |
| | Difference | -0.79 | 0.400 | [-2.66, 1.08] | -0.45 | 0.710 | [-2.84, 1.95] |
| **Pulses** | Mexico | -1.70 | 0.020 | [-3.12, -0.28] | -0.80 | 0.140 | [-1.86, 0.26] |
| | Synthetic control | 1.52 | 0.000 | [0.54, 2.49] | 1.12 | 0.030 | [0.12, 2.12] |
| | Difference | -3.22 | 0.000 | [-4.94, -1.50] | -1.92 | 0.010 | [-3.38, -0.46] |
| **Nuts** | Mexico | 1.16 | 0.000 | [0.88, 1.43] | -0.44 | 0.050 | [-0.88, 0.01] |
| | Synthetic control | 0.14 | 0.010 | [0.04, 0.24] | 0.08 | 0.040 | [0.00, 0.16] |
| | Difference | 1.01 | 0.000 | [0.72, 1.31] | -0.52 | 0.020 | [-0.97, -0.07] |
| **Animal-source food** | Mexico | 11.41 | 0.000 | [9.34, 13.48] | 8.01 | 0.000 | [6.30, 9.71] |
| | Synthetic control | 4.90 | 0.000 | [3.19, 6.61] | -1.29 | 0.370 | [-4.11, 1.54] |
| | Difference | 6.51 | 0.000 | [3.82, 9.20] | 9.30 | 0.000 | [6.00, 12.60] |
| **Meat** | Mexico | 6.81 | 0.000 | [5.73, 7.89] | 6.17 | 0.000 | [4.85, 7.49] |
| | Synthetic control | 5.59 | 0.000 | [4.46, 6.73] | 4.68 | 0.000 | [4.01, 5.34] |
| | Difference | 1.22 | 0.130 | [-0.35, 2.79] | 1.50 | 0.050 | [0.02, 2.97] |
| **Eggs** | Mexico | 1.87 | 0.000 | [1.53, 2.21] | 1.02 | 0.000 | [0.78, 1.26] |
| | Synthetic control | 0.92 | 0.000 | [0.67, 1.17] | 0.23 | 0.040 | [0.02, 0.45] |
| | Difference | 0.95 | 0.000 | [0.53, 1.37] | 0.78 | 0.000 | [0.46, 1.11] |
| **Milk** | Mexico | 2.64 | 0.000 | [1.85, 3.44] | 0.75 | 0.000 | [0.40, 1.11] |
| | Synthetic control | -1.45 | 0.050 | [-2.92, 0.03] | -1.46 | 0.000 | [-1.91, -1.02] |
| | Difference | 4.09 | 0.000 | (2.42, 5.76) | 2.22 | 0.000 | (1.65, 2.79) |

Coefficients estimated with ITSA model. The control group was estimated with synthetic controls from lower-middle and upper-middle-income countries, adjusted for supply of fat and protein (gr/capita/day) provided by the modeled food group, gross domestic product, life expectancy at birth, $CO_2$ emissions per capita, the proportion of the rural population, and primary school enrollment. 95%CI—95% confidence interval.

kcal/capita/day due to NAFTA. In the second phase after full implementation of NAFTA (2008–2018), the difference in the apparent consumption trends of nuts reflected a significant reduction, with a value of -0.52 kcal/capita/day with p = 0.02.

Trends in apparent consumption of animal-source foods during the post-1994 period show a significant increase in Mexico, with values of 11.40 kcal/capita/day (p = 0.00). The synthetic control showed a significant increase of 4.90 kcal/capita/day (p = 0.00) during the same period. The difference was 6.51 kcal/capita/day (p = 0.00). The difference in the second phase was 9.30 kcal/capita/day (p = 0.00) comparing Mexico with control group, therefore attributable to the NAFTA implementation.

Apparent consumption of meat shows a significant increase in both treatment and control. The post-1994 consumption trends were 6.81 and 5.59 kcal/capita/day (p < 0.05), and the post-2008 trends were 6.17 and 4.68 kcal/capita/day (p < 0.05) for Mexico and the synthetic control, respectively. The 1.50 kcal/capita/day (p = 0.05) difference in apparent consumption between study groups was significant in the second phase (2008–2018). There were significant increases in eggs consumption in Mexico; 1.86 and 1.02 kcal/capita/day (p < 0.05) in the 1994 and 2008 post-intervention periods. However, the synthetic control also increased by 0.92 kcal/capita/day and 0.23 kcal/capita/day (p < 0.05) in post-1994 and post-2008 periods

respectively. Differences between groups was 0.94 and 0.78 kcal/capita/day (p < 0.05) for both pos intervention periods, higher consumption in Mexico.

Finally, for milk, the trends of apparent consumption in Mexico were positive in the post-1994 and 2008 periods, with values of 2.64 and 0.75 kcal/capita/day (p < 0.05). On the other hand, the synthetic control reported negative values. As a result, the difference in the trends of apparent consumption reflected values of 4.09 and 2.22 kcal/capita/day (p < 0.05), representing a significant increase in the apparent consumption of milk in Mexico due to NAFTA.

## Discussion

We estimated changes of apparent staple food consumption (kcal/capita/day) associated with the gradual implementation of NAFTA in Mexico. To approximate the causal effect of NAFTA on the outcome of interest, we compared Mexico with a synthetic control using a multiple group ITSA model.

Our results show that when comparing Mexico's staple food consumption trends and a synthetic control (not exposed to a trade agreement), there was a significant reduction in the apparent consumption of pulses after initial implementation of NAFTA in 1994 and after full implementation in 2008. The vegetable-source foods showed a significant increase after 2008. Nuts showed a positive difference after 1994, but a negative one after 2008. On the other hand, the apparent consumption trends of animal-source foods, eggs, and milk showed significant increases between 1994 and 2008, and the meat group showed only an increase after 2008.

The relationship between the implementation of NAFTA and the increase in consumption of animal-source food is consistent with evidence from other regional trade agreements. Countries participating in the CAFTA-DR agreement significantly increased meat imports from the US: percentage changes between 2005 and 2016 were 2,153% for cattle, 1,135% for pigs, 460% for poultry, and 488% for processed meat products [34]. Between 1990 and 2007, trade liberalizations were also associated with increased imports of livestock products and poultry in Samoa and sheep meat in Fiji [35].

In 2019, an analysis based on the National Household Income and Expenditure Survey (ENIGH for its acronyms in Spanish) examined trends in food consumption based on the degree of processing during the period 1984–2016 [36]. Although this study did not focus on the effects of trade liberalization, the authors adjusted trends by household and macroeconomic variables, and the results show an increase in consumption of food of animal origin and a decrease in consumption of pulses. These results are consistent with the trends we observed.

In the absence of an experimental design, we cannot fully isolate NAFTA's impact on food consumption from other factors such as economic growth or technological development [9]. However, the inclusion of a synthetic control with balanced observable characteristics such as previous trends, gross domestic product, the proportion of the rural population, or $CO_2$ emissions provides a good approximation.

The data reported by FBS represent only the average of stocks available in a country but are not a direct indicator of individual final food consumption [26]. The data matrix does not consider correction factors for domestic food waste; especially in low- and middle-income countries, domestic food waste can be substantial [37] under this assumption, the amounts of apparent consumption could be overestimated for vegetables and underestimates for pulses and nuts [38]. However, this potential bias is present in both treatment and control, and it does not affect the validity of the estimated differences. Another limitation is that the FBS data does not include information on the type of preparation or the degree of food processing at consumption. Therefore, it is not possible to show associations between food consumption and negative health outcomes (cardiovascular disease, diabetes, or obesity) at individual level.

The results of this study should be interpreted with caution. The ideal strategy to evaluate a program is an experimental design where treatment and control groups are randomly assigned which are not plausible for policies implemented at the national level such as NAFTA. We acknowledge the limitations of using a pool of countries as control groups for comparison. The construction of a reliable synthetic control depends on the quality of the data and finding a good set of donor units can be challenging. In this study, we used data from the World Bank and FAO to create the synthetic control. Although the donor pool was made up of 55 countries, the construction of each synthetic control only considered between 3 and 7 countries that were not statistically different from Mexico in the variables used to construct this pool. In addition, using ITSA models allowed to test basic characteristics of an adequate control group: no statistical difference in the magnitude and trend pre-NAFTA between the synthetic control and Mexico. We acknowledge that even if we excluded countries with no formal trade agreements with the United States or Canada, donor units may not be completely free of any transnational trade.

Despite the limitations mentioned above, FBS estimates of apparent consumption of staple foods provide a reliable overview of the nutrition situation in Mexico [25]. Moreover, the ITSA model with multiple groups and symmetrically distributed observations over time (30 years before and 30 years after the intervention) is considered one of the most robust quasi-experimental designs for assessing the impact of large-scale interventions [30].

## Policy implications

This study provides estimates of the impact of NAFTA on apparent staple food consumption in Mexico. Ours is one of the first studies to examine the relationship between trade liberalization and changes in the trend of staple food consumption using a quasi-experimental design.

NAFTA affected food of animal- and vegetable-source consumption differently. Compared with the synthetic control, apparent consumption of animal-source foods increased, consumption of pulses decreased, while fruits and vegetables remained stable. Animal foods are a rich source of proteins with high biological value, necessary for adequate growth and human development, which is particularly important in low- and middle-income countries [39]. However, high consumption of animal-source food also negatively affects health and the environment (e.g., greenhouse gas emissions and water pollution) [40]. Shifting to a balanced diet with higher protein intake from plant-based foods such as whole grains, pulses, nuts, seeds, fresh fruits, and vegetables represents a significant step forward in improving human health and environmental sustainability [41]. Many of the resources used to produce livestock feed could be used more efficiently by growing plant foods for human consumption [42].

Presenting the results at an aggregate level does not allow us to observe the differential impact of NAFTA on different populations in Mexico (depending on the socioeconomic level or geographic area). Future studies should assess the impact of NAFTA on population groups that are particularly vulnerable to unhealthy dietary changes, such as people of low socioeconomic status [4]. This will help inform the development and evaluation of future strategies to protect population nutrition and health. There is a lack of information on fiscal strategies explicitly aimed at reducing meat consumption. However, it has been observed that specific fiscal policies, such as taxes on saturated fat, can reduce the consumption of highly processed meat products [43]. There is a need to generate evidence on the structural determinants of meat and derivative consumption and the possible strategies to promote an optimal intake in terms of quality and quantity. A key premise of free trade is that the removal of barriers leads to increased welfare and a more efficient allocation of resources [11]. However, there is evidence that trade agreements can generate negative externalities in social, environmental, food

and therefore health aspects [21,44,45], so it is important to include policy measures that strengthen local healthy food production and processing, to improve the environment and human wellbeing.

## Conclusion

Our research suggests that the implementation of NAFTA had a differential impact on the apparent consumption of staple foods when comparing Mexico and a synthetic control. NAFTA is associated with an increase in apparent consumption of animal-source food and a decrease in consumption of pulses, while there are no significant differences in consumption of fruits and vegetables. Given the environmental and health impact of high consumption of animal-source foods, it is essential to promote policies at the macro level aimed at creating a healthier food system and, at the individual level, to equip the population with tools to make more informed and sustainable food choices. Although for this study we excluded countries with trade agreements with the US or Canada to create a pool of units as controls not exposed to the program, future studies should compare countries with trade agreements similar to Mexico to analyze if changes in consumption or dietary patterns were similar.

## Supporting information

**S1 Table. Weighted values of each donor unit for the synthetic controls prediction.**
(DOCX)

**S2 Table.** a: Macroeconomic variables for Mexico and the synthetic control. b: Apparent consumption, total calories, protein, and fat (gr/capita/day) for different food groups for Mexico and the synthetic control.
(DOCX)

## Acknowledgments

We thank María Fernanda Kroker Lobos, Mauricio Hernández Fernández, Jose Luis Figueroa Oropeza, and Monica Ancira Moreno for their comments and review of the manuscript.

## Author Contributions

**Conceptualization:** Néstor A. Sánchez-Ortiz, Mishel Unar-Munguía, Sergio Bautista-Arredondo, Teresa Shamah-Levy, M. Arantxa Colchero.

**Formal analysis:** Néstor A. Sánchez-Ortiz, M. Arantxa Colchero.

**Investigation:** Néstor A. Sánchez-Ortiz, Sergio Bautista-Arredondo, Teresa Shamah-Levy, M. Arantxa Colchero.

**Methodology:** Néstor A. Sánchez-Ortiz, Mishel Unar-Munguía, Sergio Bautista-Arredondo, M. Arantxa Colchero.

**Supervision:** M. Arantxa Colchero.

**Writing – original draft:** Néstor A. Sánchez-Ortiz.

**Writing – review & editing:** Néstor A. Sánchez-Ortiz, Mishel Unar-Munguía, Sergio Bautista-Arredondo, Teresa Shamah-Levy, M. Arantxa Colchero.

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
