## [Decision Letter · Decision Letter 0]

16 Aug 2022

PGPH-D-22-01031

Changes in apparent consumption of staple food in Mexico associated with the gradual implementation of the NAFTA

Dear Dr. Colchero,

Thank you for submitting your manuscript to PLOS Global Public Health. After careful consideration, we feel that it has merit but does not fully meet PLOS Global Public Health’s publication criteria as it currently stands. Therefore, we invite you to submit a revised version of the manuscript that addresses the points raised during the review process.

We look forward to receiving your revised manuscript.

Kind regards,

Edna N Bosire, Ph. D

Academic Editor

Journal Requirements:

2. Please update the 'Competing Interests' statement, including any COIs declared by your co-authors. If you have no competing interests to declare, please state "The authors have declared that no competing interests exist"

3. Please provide separate figure files in .tif or .eps format.

Additional Editor Comments (if provided):

Dear Authors,

Thank you for submitting your manuscript titled "Changes in apparent consumption of staple food in Mexico associated with the gradual

implementation of the NAFTA". This is a well written manuscript and will be of interest to PlosGPH readers.

We have had challenges to get reviewers for this manuscript. However, one reviewer has provided his comments and concerns over the methodological aspect of your work. You should consider these reviews to help improve your work.

Please revise and send back to us for another round of reviews. We may also have to get other additional reviewers for the second round of reviews.

Minor reviews: Please re-read your manuscript and correct any inconsistencies e.g., use of quotation marks. Also see some spellings e.g. line 136 and 234 the word "therefore" is incomplete.

Reviewers' comments:

Reviewer's Responses to Questions

**Comments to the Author**

1. Does this manuscript meet PLOS Global Public Health’s publication criteria? Is the manuscript technically sound, and do the data support the conclusions? The manuscript must describe methodologically and ethically rigorous research with conclusions that are appropriately drawn based on the data presented.

Reviewer #1: Partly

2. Has the statistical analysis been performed appropriately and rigorously?

Reviewer #1: No

3. Have the authors made all data underlying the findings in their manuscript fully available (please refer to the Data Availability Statement at the start of the manuscript PDF file)?

Reviewer #1: Yes

4. Is the manuscript presented in an intelligible fashion and written in standard English?

Reviewer #1: Yes

5. Review Comments to the Author

Reviewer #1: Thank you for the opportunity to review this article. The authors assessed the "relationship between trade liberalization and changes in the trend of staple food consumption using a quasi-experimental design". The articles is presented well and gives insight into an important subject, which adds value to the field. However, I have several concerns with the methodological approach which the authors employed.

The authors modeled splines in three periods, before 1994, 1994-2007, and from 2008. Although there is possibly good rationale for using 1994 and 2008 as cut-offs, I question if this does not create artificial inflection points and thereby enforcing a linear relationship where one might not exist. The raw data for both the intervention and control show that there are non-linear patterns within the three periods while the predicted line is linear. An example of this is the apparent consumption of animal-source foods. These show an increase in both Mexico and the synthetic controls before 1994. However, the rise was steeper for Mexico than for the control, and both experienced a decline before 1994. The regression slopes to do account for this steep increase in Mexico, but instead show a steeper increase between 1994 and 2008 and do not capture the observable steep increase in the control in the same period. I question whether the model fits the data well and this can be said for other plots as well. This for me is a major drawback of the manuscript and I suspect that a different model may lead to different conclusions.

In addition, I made other comments that could assist in improving the manuscript:

1. Line 183-185: What is the implication of excluding the non-parallel groups? I’m struggling to follow what is presented in table 2. The text also does not give details of this comparison? Are you presenting regression coefficients of apparent consumption for each of the categories, comparing Mexico to the synthetic controls?

Table 1 and S2: This may be my ignorance, but how come the school enrollment is above 100%?

Table 2: The table can do with a more self-explanatory legend and an explanation of what the values in the table are.

Table S2: The table is presenting two different sets of data. A comparison of key indicators between Mexico and the synthetic controls derived for the food categories. Secondly, it shows a comparison of apparent consumption in the different food categories between Mexico and the synthetic controls. Perhaps the manuscript might benefit from separating the table into two. It took me a while to realise what was happening

6. PLOS authors have the option to publish the peer review history of their article (what does this mean?). If published, this will include your full peer review and any attached files.

**Do you want your identity to be public for this peer review?** For information about this choice, including consent withdrawal, please see our Privacy Policy.

Reviewer #1: **Yes: **Lukhanyo H. Nyati

---

## [Decision Letter · Decision Letter 1]

17 Oct 2022

PGPH-D-22-01031R1

Changes in apparent consumption of staple food in Mexico associated with the gradual implementation of the NAFTA

Dear Dr. Colchero,

Thank you for submitting your manuscript to PLOS Global Public Health. After careful consideration, we feel that it has merit but does not fully meet PLOS Global Public Health’s publication criteria as it currently stands. Therefore, we invite you to submit a revised version of the manuscript that addresses the points raised during the review process.

We look forward to receiving your revised manuscript.

Kind regards,

Edna N Bosire, Ph. D

Academic Editor

Journal Requirements:

Additional Editor Comments (if provided):

Dear Authors,

Thank you for responding to reviewers comments. One reviewer has raised important questions in your manuscript that you may have to address before we can proceed with your manuscript.

We look forward to your revised manuscript.

Reviewers' comments:

Reviewer's Responses to Questions

**Comments to the Author**

1. If the authors have adequately addressed your comments raised in a previous round of review and you feel that this manuscript is now acceptable for publication, you may indicate that here to bypass the “Comments to the Author” section, enter your conflict of interest statement in the “Confidential to Editor” section, and submit your "Accept" recommendation.

Reviewer #2: All comments have been addressed

Reviewer #3: (No Response)

2. Does this manuscript meet PLOS Global Public Health’s publication criteria? Is the manuscript technically sound, and do the data support the conclusions? The manuscript must describe methodologically and ethically rigorous research with conclusions that are appropriately drawn based on the data presented.

Reviewer #2: Yes

Reviewer #3: Yes

3. Has the statistical analysis been performed appropriately and rigorously?

Reviewer #2: Yes

Reviewer #3: Yes

4. Have the authors made all data underlying the findings in their manuscript fully available (please refer to the Data Availability Statement at the start of the manuscript PDF file)?

Reviewer #2: Yes

Reviewer #3: Yes

5. Is the manuscript presented in an intelligible fashion and written in standard English?

Reviewer #2: Yes

Reviewer #3: Yes

6. Review Comments to the Author

Reviewer #2: (No Response)

Reviewer #3: The study attempts to answer an important research topic: Evaluate the association between the gradual implementation of NAFTA and changes in apparent consumption of staple foods in Mexico, which is especially pertinent given the health implications of these consumptions patterns, and in congruence with the WHO’s emphasis on the importance of the balance between “trade with health” as quoted by authors.

There are limitations to using the synthetic control model and to the conclusions that can be made based on it being a 'control'. These need to be objectively and adequately discussed. An important paper and calls for further research, such as that comparing consumption of dietary patterns with other countries that have trade deals with the US and Canada, or countries that are part of other trade deals (not US and Canada), and compare and contrast those. Interesting work.

7. PLOS authors have the option to publish the peer review history of their article (what does this mean?). If published, this will include your full peer review and any attached files.

**Do you want your identity to be public for this peer review?** For information about this choice, including consent withdrawal, please see our Privacy Policy.

Reviewer #2: **Yes: **Anisha Adhikari

Reviewer #3: No

---

## [Editor Report · Decision Letter 2]

3 Nov 2022

Changes in apparent consumption of staple food in Mexico associated with the gradual implementation of the NAFTA

PGPH-D-22-01031R2

Dear Dr. Colchero,

We are pleased to inform you that your manuscript 'Changes in apparent consumption of staple food in Mexico associated with the gradual implementation of the NAFTA' has been provisionally accepted for publication in PLOS Global Public Health.

Best regards,

Edna N Bosire, Ph. D

Academic Editor